# Efficient and Accurate 3-D Numerical Modelling of Landslide Tsunami

**Guodong Li [1,*]**, **Guoding Chen [1,2]**, **Pengfeng Li [1]** and **Haixiao Jing [1]**

[1]  State Key Laboratory of Eco-hydraulics in Northwest Arid Region of China, Xi'an University of Technology, Xi'an 710048, China; cgdwork@hhu.edu.cn (G.C.); lipengfeng1994@163.com (P.L.); jinghx@xaut.edu.cn (H.J.)

[2]  State Key Laboratory of Hydrology-Water Resources and Hydraulic Engineering, and College of Hydrology and Water Resources, Hohai University, Nanjing 210098, China

\*  Correspondence: gdli2008@xaut.edu.cn

**Abstract:** High-speed and accurate simulations of landslide-generated tsunamis are of great importance for the understanding of generation and propagation of water waves and for prediction of these natural disasters. A three-dimensional numerical model, based on Reynolds-averaged Navier–Stokes equations, is developed to simulate the landslide-generated tsunami. Available experiment data is used to validate the numerical model and to investigate the scale effect of numerical model according to the Froude similarity criterion. Based on grid convergence index (GCI) analysis, fourteen cases are arranged to study the sensitivity of numerical results to mesh resolution. Results show that numerical results are more sensitive to mesh resolution in near field than that in the propagation field. Nonuniform meshes can be used to balance the computational efficiency and accuracy. A mesh generation strategy is proposed and validated, achieving an accurate prediction and nearly 22 times reduction of computational cost. Further, this strategy of mesh generation is applied to simulate the Laxiwa Reservoir landslide tsunami. The results of this study provide an important guide for the establishment of a numerical model of the real-world problem of landslide tsunami.

**Keywords:** landslide tsunami; wave model; scale effect; wave region division; mesh generation schemes

## 1. Introduction

Landslides are a common type of geological disaster. Tsunamis caused by landslides damage buildings and injure residents [1]. In the Vaiont Landslide of 1963, a mass of approximately 270M m$^3$ collapsed into a reservoir, generating a wave that overtopped the dam and hit the town of Longarone and other villages. Nearly 2000 people lost their lives in this disaster [2]. Recently, owing to climate change and the frequent occurrence of extreme weather, landslide disasters present a serious challenge. Thus, investigating landslide tsunamis is ever more necessary.

For a better understanding of landslide tsunami, many simplified laboratory experiments have been performed [3,4]. The experimental setup often comprises a channel (e.g., 10–20 m), slide, and slide releaser. Slide width has been used to determine two-dimensional or three-dimensional (3-D) issues [5]. Many researches have used stylus and high-speed cameras to capture the shapes of waves [6–8], and particle image velocimetry is used to capture the flow field [9–11]. To make the experiment more realistic, the experimental scale has been further amplified. Liu et al. [12] launched a large-scale 3-D experiment at the 100-m level. In Wang's [13] experiment, realistic topography was employed rather than rectangular channels, but the slide was considered a rigid block of regular shape. To a certain extent, the data from physical experiments have helped scholars better understand the processes of

landslide tsunamis. However, because of the complex topography and large scale of real engineering, full-scale experiments have nearly always been impossible. The difficulty and cost of the experiment make scholars more willing to use mathematical models to study landslide-tsunami disasters [14].

In recent decades, the mathematical models for landslide tsunamis have been widely researched and accepted. Studies have attempted to develop mathematical models and algorithmic innovations [14–16] to study and explain the phenomenon of landslide tsunamis [5]. In recent years, numerical models have based on the shallow water equation [4,17], Boussinesq-type model [18,19], and Navier–Stokes equation [16,20–22] have been used. Wang et al. [23] proposed a new method for considering dry–wet boundary treatment of a slider into water, coupled with the shallow water equation of the wave. Lo et al. [24] compared the differences between the three wave models. For free surface flows, Ma et al. [25] provided a shock-capturing non-hydrostatic model for fully dispersive surface wave processes. Gallerano et al. [26] proposed a new three-dimensional finite-volume non-hydrostatic shock-capturing model for free surface flow. More recently, the smooth-particle method has also been widely used in landslide-tsunami research [27–29]. Due to the superiority of mathematical models, actual engineering applications have also been attempted by researchers [30,31]. Ataie et al. [32] derived the 2-D (two-dimensional) 4th-order Boussinesq equation, verified by physical experiments and applied to the Maku and Shafa–Roud reservoir areas. Yin et al. [20] used the Flow-3D package to simulate the landslide disaster in the Three Gorges Reservoir, China. At the same location as in Reference [20], Huang et al. [4] coupled the dimensionless formula and Boussinesq-type to calculate and analyze two representative examples from landslides. Generally, the mathematical models derived in previous literatures are mostly based on indoor experimental scales, focusing on the study of water wave models [14]. The application of the landslide-tsunami model on a real-world scale which involves complex topography and slide surface is still inadequate.

Lacking a landslide-tsunami disaster record, the numerical models for landslide-tsunami disasters in the real world are often verified via laboratory experimental data [14]. However, hydraulic scale modelling involves scale effects [33,34], which should be considered in the scale transformation of actual application. Scale effects were primarily attributed to the impact crater formation, the air entrainment and detrainment, and the turbulent boundary layer as a function of surface tension and fluid viscosity. These effects reduce the relative wave amplitude and the wave attenuation as compared with reference experiments [33]. Inevitably, the scale effect of the numerical model should be validated prior to its application. Moreover, as a disaster requirement, computational efficiency is considered an important indicator in the generation and propagation of landslide tsunamis, involving large calculation areas. Although uniform [20] or nonuniform meshes [35] have been adopted, study of sensitivity analyses of different domains and mesh sizes is also significant for the mesh-generation scheme. Furthermore, the landslide-tsunami-type problem itself has obvious far- and near-field calculation characteristics (e.g., the shape and volume of the landslide have a greater impact on the near-field wave feature [24]). Thus, a reasonable mesh-generation strategy based on sensitivity analysis is needed to balance computational efficiency and accuracy.

In this work, the study focuses on the scale effect and mesh generation of the numerical model for landslide tsunamis. Thus, the numerical method is introduced in the next section. The proposed method is validated by previous experiment data and other numerical results. Simultaneously, scale effect is investigated in different length scales of the numerical model. On a scale with practical engineering, mesh dependence is analyzed under the global uniform mesh and the mesh-generation scheme is proposed. In Section 5, the numerical model and mesh-generation scheme are applied to landslide tsunamis in the Laxiwa Reservoir, China. Finally, the conclusions are drawn in Section 6.

## 2. Numerical Method

Landslide tsunamis are a major challenge for computational fluid dynamics (CFD), owing to the multi-phases and multi-materials [5]. Numerical models are required to accurately calculate high-speed slide impacts and complex dry–wet boundary transformations [23]. In this study, the available

CFD package (i.e., Flow-3D) based on the 3-D Reynolds-averaged Navier–Stokes (RANS) equations is used to calculate the flow-field and the renormalization group turbulent model is used for the closure of RANS equations. Note that the governing equations are described under the Cartesian coordinate system with the z-axis being vertical and the gravity direction being vertically downward. The RNG(Re-normalization group) model in Flow-3D can give the following equation:

$$\frac{\partial k_T}{\partial t} + \frac{1}{V_F}\left(uA_x\frac{\partial k_T}{\partial x} + vA_y\frac{\partial k_T}{\partial y} + wA_z\frac{\partial k_T}{\partial z}\right) = P_T + G_T + Diff_T - \varepsilon_T \tag{1}$$

where $k_T$ denotes turbulent kinetic energy; $P_T$ and $G_T$ are the turbulent kinetic energy term and buoyancy term, respectively; $Diff_T$ denotes the diffusion term, and $\varepsilon_T$ denotes the turbulent dissipation term. $(u, v, w)$ are the components of velocity vector in the three directions $(x, y, z)$. $V_F$ represents the fluid volume fraction in each mesh unit. In order to improve the efficiency of calculation, the fluid area fraction $A_x, A_y, A_z$ (in three direction) are used for each calculation unit during the calculation. Note that the intermixing of landslide material with water is neglected in this study so that the water density is considered as constant for simplification ($1000 \text{ kg/m}^3$ is used here).

The volume-of-fluid (VOF) method, a relatively mature method for free-surface tracking [36], is adopted at the free boundary to capture the rapidly changing water surface. Under the VOF method framework, the fluid (water in this paper) volume fraction of each calculation unit is treated as a volume function which can effectively distinguish different phases [37]. This volume is then transported directly at local speeds without the transport equation. The superior performance of the VOF method guarantees absolute conservation of the fluid volume. The VOF method has been widely used, and its reliability has been proven [20,38,39]. For efficiency, the Tru–VOF method [40,41] is used to ignore the gas unit. Compared with traditional VOF, the calculation convergence time can be shortened and the free-surface change can be described more accurately.

Landslide tsunamis involve landslide detachment and water-body coupling [42]. Thus, the correct fluid–solid coupling method is critical to this problem. Flow-3D provides two ways to update the motion state of the slide block: prescribed motion and couple motion. Prescribed motion is set in case that the experimental data is available (as in the validation part in this paper), while couple motion is often used for prediction. The general moving object (GMO) model is used to describe the motion of the moving slide [20,32,43]. GMO is a rigid body under any type of physical motion, which is either dynamically coupled with fluid flow or is user prescribed. It can move with six degrees of freedom (DOF) or rotate about a fixed point or a fixed axis. The geometric model of the new moment is calculated by the mesh encroachment at each time step. The deformation of the landslide and collision with the slope are ignored. The combined GMO method together with the fluid governing equation can be used to solve the fluid–solid coupling problem involved in a landslide tsunami. The model we selected has been successfully used to simulate impulse wave generated by landslides [20,44]. The computational meshes includes not only the watershed but also the free space for the development of water waves so that the total mesh volume can be larger than the initial water volume. Given that fluid volume is a function during the whole process, the wet boundary is updated with the the progress of calculation. Thus, the wet–dry transformation is achieved. More details of the physical and numerical model are described in Flow Science [20].

Types of landslides have strong regional differences, which are related to land cover, land use, and climate condition, etc. [45,46]. As previously reported, rigid slides produce more turbulence as well as greater wave runup and wave height. The bluff body effect of rigid landslides seems to significantly influence the wave height and runup [47]. In our study, the mesh-generation schemes and efficient and accurate numerical model establishment are focused. It needs to be emphasized that the landslide block deformation and fragmentation are not considered and that the block itself is treated as a rigid body. In addition, neglecting the deformation is consistent with the verification experiment selected in the following sections. It also has to be emphasized that the air portion of each calculation unit is not considered in the calculation process for simplification.

## 3. Model Validation and Scale Effect

For validation, the proposed model (Section 2) is used to replicate the experiments of Heinrich [7] and Heller et al. [5], which describe subaerial and submarine landslides, respectively. Current solution is also compared to other numerical results. The scale effect of the numerical model is also investigated and discussed in this section.

### 3.1. Validation: Subaerial and Submarine Landslide

In Heinrich's experiment [7], a submarine rigid slide was used to generate waves. The rigid wedge with a density of 2000 kg/m$^3$ moved down a ramp with a slope of 45°, and detailed experimental data was recorded. Thereafter, the results of the physical experiment were widely used in other studies to validate their numerical results [17,29,48]. The schematic of the experiment is depicted in Figure 1a, and the corresponding parameters are given in Table 1. The velocity of the slide used in the calculation is recorded via experiments. The upper boundary ($z_{max}$) is the free-surface boundary, and the remaining boundaries are solid boundaries. The calculation domain is uniformly meshed by orthogonal hexahedrons with a side length of 1.5 cm, and the total mesh number is 2.96 M.

The schematic of subaerial experiment is depicted in Figure 1b. The experiment is performed on a 2-D channel and a 3-D basin for two different scenarios reported in Reference [5]. In this paper, we perform the simulation in 2-D for scenario 1 reported in Reference [5], where the still water depth is 0.24 m and the impact results in an impulsive solitary wave. The important parameters of the experiment are summarized in Table 2. The detailed description of the experiment can be found in Reference [5]. The mesh size is set to 1.5 cm, resulting in 3.5M total mesh number. The boundary conditions are set to the same as that of the submarine scenario.

The free surface at t = 1.5 s and t = 3.0 s are compared with submarine experimental data of Heinrich [7] and shown in Figure 2. The Nasa-vof2D results which have been reported in Heinrich [7] are also compared with the current solution. It can be found in Figure 2 that the wave features at these two moments are better resolved in this paper, especially at 1.5 s. The RANS model can better capture the dispersive effect of the wave (due to the strong disturbance of the water surface as stated before [49]). Note that the generated waves have not yet propagated to the right extremity of the tank at 3 s, and thus, the wave reflections can be neglected.

In Figure 3, the variations of water surface elevation are extracted and quantitatively compared with the experimental data [5] (serves as a benchmark herein) and other numerical results. The overall agreement is good, particularly the location and propagation velocity of the primary wave. The wave amplitude can be captured more accurately via local mesh refinement. Both submarine and subaerial scenarios are well reproduced by the proposed model, demonstrating its capabilities in rigid landslide tsunamis.

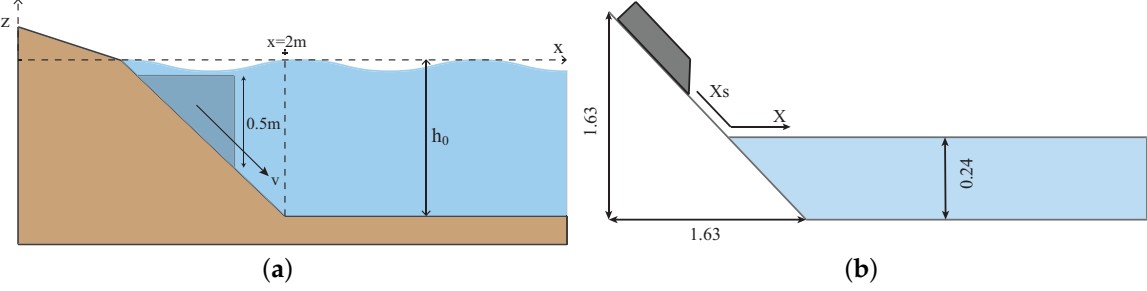

**Figure 1.** Experimental setup. (**a**) Submarine landslide of Heinrich [7]. (**b**) Subaerial landslide of Heller et al. [5]

**Table 1.** Summary of important parameters in the Heinrich [7] experiment (Figure 1a).

| Parameter | Value |
| --- | --- |
| slide length (two right angle sides) | 0.5 m |
| slide width | 0.55 m |
| slide mass | 140 kg |
| still water depth ($h_0$) | 1 m |
| water density ($\rho$) | 1000 kg/m$^3$ |
| initial slide position | 1 cm below the undisturbed free surface |
| slope inclination | 45° |
| channel width | 0.55 m |
| channel length | 20 m |
| wave gauges position | x = 4 m, x = 8 m, x = 12 m |

**Table 2.** Summary of important parameters in the Heller et al. [5] experiment (Figure 1b).

| Parameter | Value |
| --- | --- |
| slide length ($l_s$) | 0.599 m |
| slide width (w) | 0.577 m |
| slide thickness (s) | 0.12 m |
| slide density ($\rho_s$) | 1540 kg/m$^3$ |
| slide mass ($m_s$) | 60.14 kg |
| still water depth ($h_0$) | 0.24 m |
| slide front initial position ($x_s$) | −0.55 m |
| water density ($\rho$) | 1000 kg/m$^3$ |
| channel width | 0.6 m |
| channel length | 24 m |
| channel height | 1.5 m |
| relative wave probe distances ($x/h_0$) | 3.0, 5.0, 7.5, 10.0, 15.0, 22.5, 35.0 |

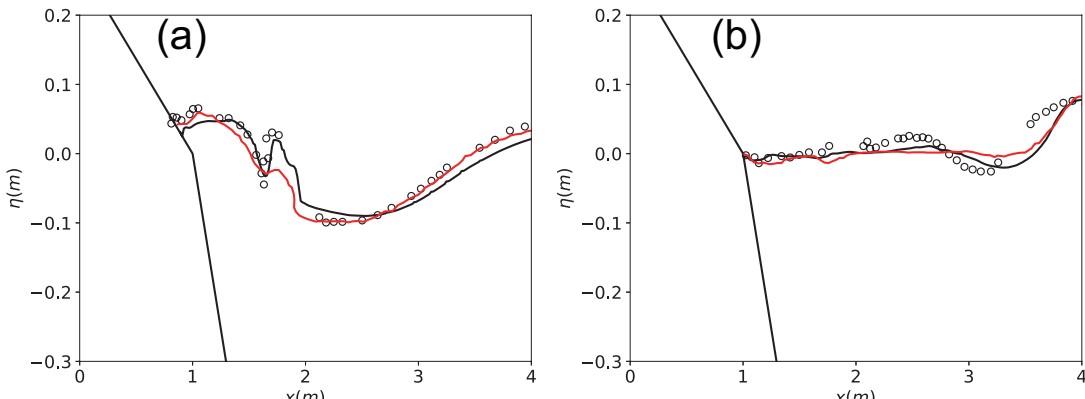

**Figure 2.** Water surface elevation $\eta$ at different time. (**a**) t = 1.5 s; (**b**) t = 3 s: experimental data (circle), current solution (black), and Nasa-vof2D results (red) of Heinrich [7].

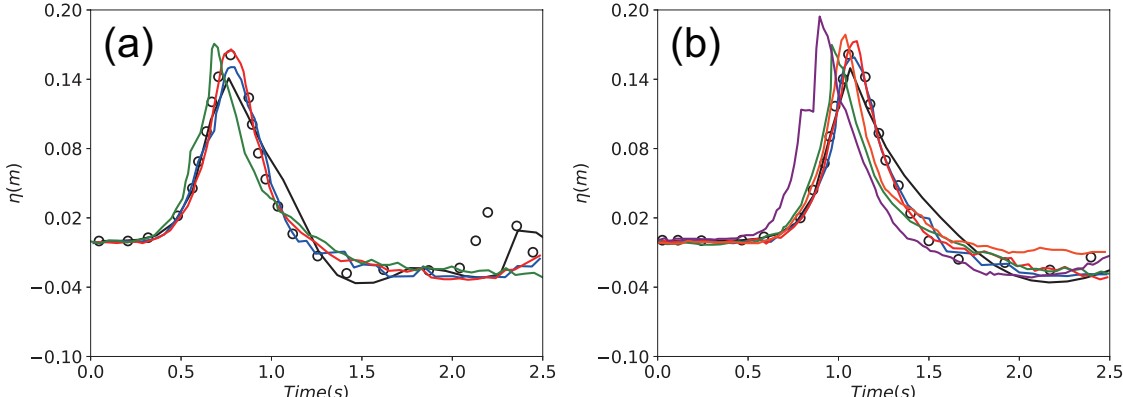

**Figure 3.** Water surface elevation $\eta$ varying with time at different positions. (**a**) x = 1.2 m and (**b**) x = 1.8 m correspond to the relative positions x/h = 5.0 and 7.5 in Heller et al. [5]: experimental data (circle), current solution (black), ISPH results (ISPH(a), blue; ISPH(b), red) of Yeylaghi et al. [29], DEM-SPH results (green) [49], and DualSHPysics results (DualSPHysics(a), orange; DualSPHysics(b), purple) of Heller et al. [5]. DualSPHysics(a) results are obtained using a reduced slide front-impact velocity of 1.32 m/s, and DualSPHysics(b) results are obtained using an unreduced slide front-impact velocity of 2.43 m/s; ISPH(a) results are obtained with particle resolution of 0.01 m, and ISPH(b) results are obtained with particle resolution of 0.005 m. The particle resolution in DualSPHysics is 0.01 m.

### 3.2. Model Scale Effect

Due to the good agreement between the model and experimental data in Heinrich [7], the scale effect of the numerical model is investigated under the Froude similarity criterion. The established geometric model is magnified 100 times corresponding to the geometric scale ($\lambda_l = 0.01$), which is near the size of the actual project. Under this criterion, $\lambda_u$ and $\lambda_t$ can be calculated as $\lambda_u = (\lambda_l)^{1/2} = 0.1$ and $\lambda_t = (\lambda_l)^{1/2} = 0.1$. Based on the value of the scale, a large-scale geometric model is established in the numerical model after modifying the parameter in Table 1. The number of meshes maintains a size of 1.5 m. For subsequent comparison with the experiment, the large-scale numerical results are minified according to the same criterion.

The calculation results shown in Figure 4 reveal that the slide ends after 1 s. At the initial moment, the landslide accelerates into the water and the water body receives the impact squeeze near the bottom of the slope. The water body sinks, the free-liquid surface drops to its lowest, and the first wave is formed. The initial wave surge and wet or dry boundary in the near field (NF) are accurately calculated by the proposed numerical model. This calculation is similar to the numerical results of other literatures [7,17,29,48]. Figure 5 shows that the wave feature calculated in the propagation stage and the computed wave height agree well with the experimental data result from the wave gauges in the propagation field (PF).

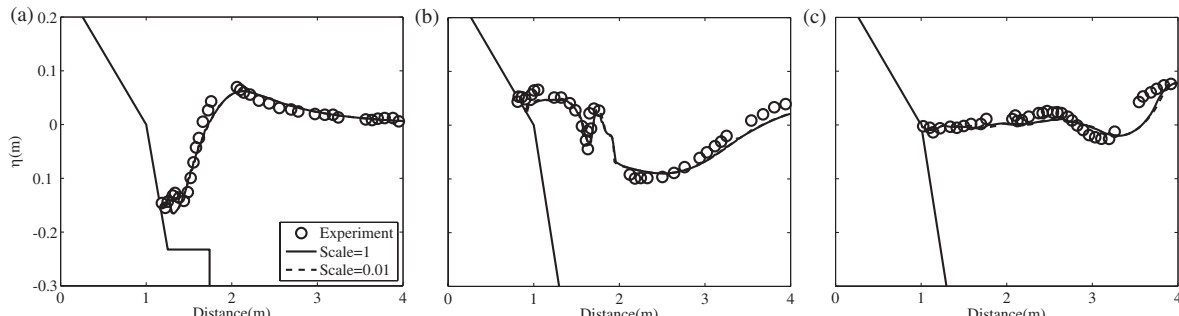

**Figure 4.** Comparison between numerical and experimental wave profiles in the near field: (**a**) t = 0.5 s; (**b**) t = 1.5 s; and (**c**) t = 3 s.

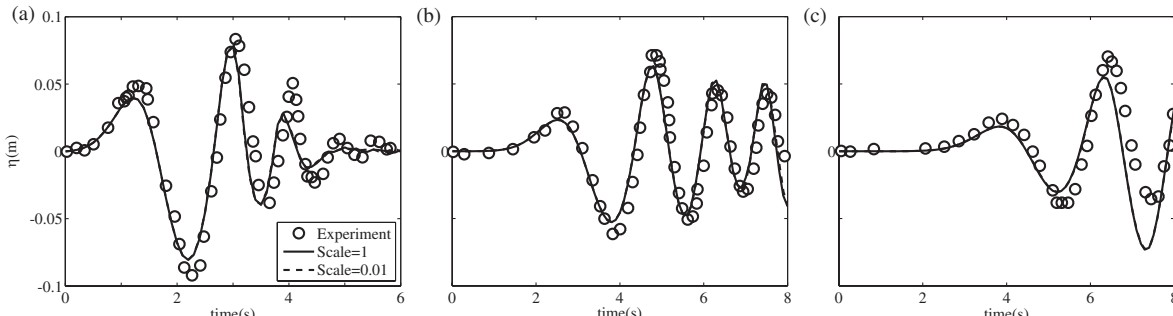

**Figure 5.** Wave gauges: Comparison between experimental and computed waves in the propagation field: (**a**) x = 4 m; (**b**) x = 8 m; and (**c**) x = 12 m.

Scale effects arise mainly as a consequence of surface tension and fluid viscosity, as reported by Heller et al. [33]. Viscosity and frequency dispersion may have a significant effect on the impulse-wave height attenuation. The presence of air was found to be responsible for scale effects in the wave-generation zone [33]. It is worth noting that the viscosity and frequency dispersion have been considered in our RANS model. Although the air is neglected in each calculation unit, it has been demonstrated to have minimal effect on the numerical results. Due to its high computational cost, the air portion is not considered as discussed above. Compared with the results of the model with the geometric scales, $\lambda_l = 1$ and $\lambda_l = 0.01$, it is found that the simulation values nearly coincide, indicating that the numerical model of the larger scale is still calculated accurately. The comparison between the numerical model at two different scales and the experiment shows that the numerical model does not have a scale effect after amplification. This indicates that the large-scale landslide problems of real-world engineering can be reasonably simulated with the proposed numerical model.

## 4. Mesh Strategy and Application

### 4.1. Numerical Cases

The landslide impulsive wave problem is complicated because the process involves the formation of impulsive waves, or jet flows, when the landslide body enters the water. The leading wave (i.e., original wave) generates and propagates immediately, and the wave energy attenuates in the later stages of propagation [4]. During calculation, a check of grid independence is necessary before determining mesh size [29,50]. When the global uniform mesh is used, although the meshing is simple, it results in extremely low computational efficiency while improving accuracy. However, the coarse mesh results in precision loss. Therefore, to balance the calculation accuracy and efficiency of the numerical model, different mesh precisions should be given in different wave areas to improve calculation speed while guaranteeing the calculation accuracy over the all-wave domain. Therefore, this section is based on the proposed numerical model to study the degree of dependence between mesh size and numerical result, focusing on the difference in free-surface capture. Therefore, the runs should be designed by global uniform mesh size and tested with the same simulation: Heinrich's.

Prior to the implementation of these cases, the grid convergence index (GCI) is necessary to be computed from the numerical-model mesh-refinement method [51,52]. Turbulence kinetic energy (TKE), which varies with simulation time because of landslide movement and wave propagation [5,8], is chosen to validate calculation convergence with a minimum four-mesh resolution. When the total amount of water is constant, the mass-average TKE is used instead. For full development of the flow field, the end of moment during the simulation is used for comparison. The factor of safety, $F_s$, is recommend in this solution with four global uniform mesh sizes at 1.5 m, 2 m, 2.5 m, and 3 m. Details of the GCI calculation are presented in Table 3. Computational results show that the GCI reduces from 17.18% to 1.94% via mesh refinement. The GCI value under a 1.5 m mesh indicates that the solutions

are well within the asymptotic range of convergence, and there is little necessary for further refining the mesh size. Based on the above analysis, 14 cases with a minimum mesh size at 1.5 m and maximum at 8 m are presented in Table 4.

**Table 3.** Grid convergence index (GCI) calculation.

| Mesh Size (m) | $r(s_1/s_2)$ | MATKE $(m^2 \cdot kg^{-1}/s^2)$ | Relative Error ($\varepsilon$) | GCI (%) |
|---|---|---|---|---|
| 3.0 | — | 0.003621 | — | — |
| 2.5 | 1.2 | 0.003854 | 0.06046 | 17.18 |
| 2 | 1.25 | 0.003976 | 0.03068 | 6.82 |
| 1.5 | 1.33 | 0.004024 | 0.01193 | 1.94 |

**Table 4.** Mesh size and total mesh number.

| Case | Mesh Size (m) | Number of Mesh |
|---|---|---|
| 1 | 1.5 | 2,960,000 |
| 2 | 2 | 1,260,000 |
| 3 | 2.5 | 633,600 |
| 4 | 3 | 360,000 |
| 5 | 3.5 | 233,376 |
| 6 | 4 | 157,500 |
| 7 | 4.5 | 107,892 |
| 8 | 5 | 79,200 |
| 9 | 5.5 | 60,060 |
| 10 | 6 | 45,000 |
| 11 | 6.5 | 33,264 |
| 12 | 7 | 29,104 |
| 13 | 7.5 | 22,400 |
| 14 | 8 | 19,740 |

*4.2. Numerical Result and Mesh-Generation Scheme*

In this simulation, the domain of the calculation is 1500 m in the x direction and is divided into NF and PF values [11,53,54]. Calculation results of free-surface levels of different mesh sizes in different computational domains are shown in Figures 6 and 7. The numerical values in cases 2~14 are compared with those of case 1 because the numerical result under the mesh size of 1.5 m is considered as the most accurate result. Figure 6 shows that the wave calculation of the NF is evidently dependent on the mesh size and that the ability to capture wave troughs in an NF decreases considerably when mesh size increases. When the mesh size increases to 8 m, the calculated differences between the troughs in Figure 6a,b are 7.5 m and 9 m, respectively, and the relative errors with the accurate values are 54.6% and 49.0%, respectively. The difference in peak calculations of the NF is not as significant as troughs. They are 3.1 m and 2 m, respectively, with relative errors of 54.5% and 46.3%.

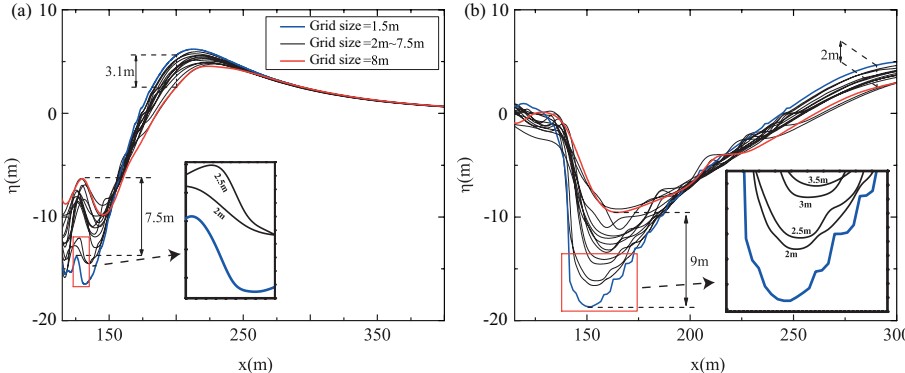

**Figure 6.** Calculation results of uniform mesh of different sizes in the near field: (**a**) t = 5 s; (**b**) t = 10 s.

After the leading wave is formed, the landslide stops moving and the resulting swell begins to propagate forward. It can be seen from Figure 7 that, after the wave enters the propagation stage, the wave calculated difference of the PF is smaller than the NF under different mesh sizes. Whether or not the peak or trough calculated difference is within 2 m, the relative error is less than 25%.

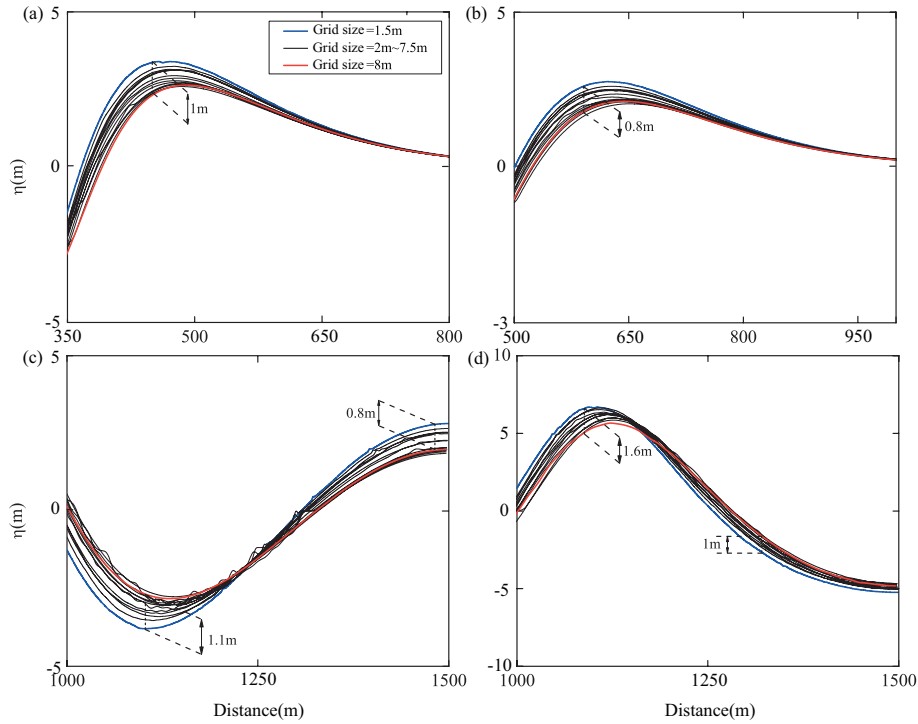

**Figure 7.** Calculation results of uniform mesh of different sizes in the propagation field: (**a**) t = 15 s; (**b**) t = 20 s; (**c**) t = 50 s; and (**d**) t = 60 s.

Overall, the results show that, with the coarsening of the calculated mesh, the ability of the numerical model to capture the free surface will be reduced and the degree of decline will be different in each computational domain. From the evident degrees of difference, the difference in NF is the most evident and the PF is not evident. The capture capacity of the coarse mesh in the NF is evidently insufficient. This difference is weakened in the PF. From the perspective of differences in peaks and troughs, the difference in trough calculations is more prominent. Thus, as the size of the mesh increases, the intensity of the impulsive wave of the simulation decreases and the ability of the model to capture peaks and troughs decreases. In other words, the calculated wave tends to flatten. Based on this calculation characteristic concerning mesh size, to give a reasonable mesh generation scheme, coarse mesh should be avoided in the NF and the fine mesh should be applied to

improve the calculation accuracy. In the PF, coarse mesh can be used to simultaneously improve the calculation speed.

To accurately and promptly obtain numerical simulation results, mesh schemes with different sizes in different calculation areas are proposed to apply to the original simulation as follows. Mesh scheme 1: a 1.5-m mesh size is used in the near field (0–400 m), mesh sizes of 3 m and 6 m are used in the PF of 400–1000 m and 1000–1500 m, respectively, and the total mesh number is 960,000. Mesh scheme 2: a 1.5-m mesh size is used in the near field (0–200 m), mesh sizes of 3 m and 6 m are used in the PF of 200–500 m and 500–1500 m respectively, and the total mesh number is 490,000. The calculation conditions are the same except for the mesh. The CPU model is Intel i7-4790, and the main frequency is 3.6 GHz. The calculation time of different mesh schemes is listed in Table 5.

The computational domain and selected moment are exactly the same as they are in Figures 6 and 7. Comparing Figure 8a,b, it can be found that, although the mesh size of mesh scheme 1 and the global uniform mesh are the same (i.e., both mesh sizes are 3 m) in this calculation area, the calculation accuracy of mesh scheme 1 is considerably improved. Similarly, the calculation accuracy of a mesh size of 6 m, used by mesh scheme 1 in the calculation area in Figure 8c,d, is better than the global uniform mesh size of 6 m. The landslide-tsunami problem can be accurately simulated under mesh scheme 1, and the same calculation accuracy as the original fine global uniform mesh (similar to the 1.5-m global-uniform mesh) can be largely achieved. From the perspective of computational efficiency, the mesh number is reduced by 68% and the calculation speed is increased 4.1 times compared with fine global uniform mesh and this progress becomes more capable as the computational area increases. However, the results also verify that the main sources of calculation error comes from inaccurate capturing of the wave characteristics in the near field. When the mesh of this domain is partially refined under mesh scheme 1, other computational domains still have better calculation accuracy under the coarse mesh. Alternatively, the mesh scheme with local refinement can help the coarse-mesh domain achieve an asymptotic range of convergence.

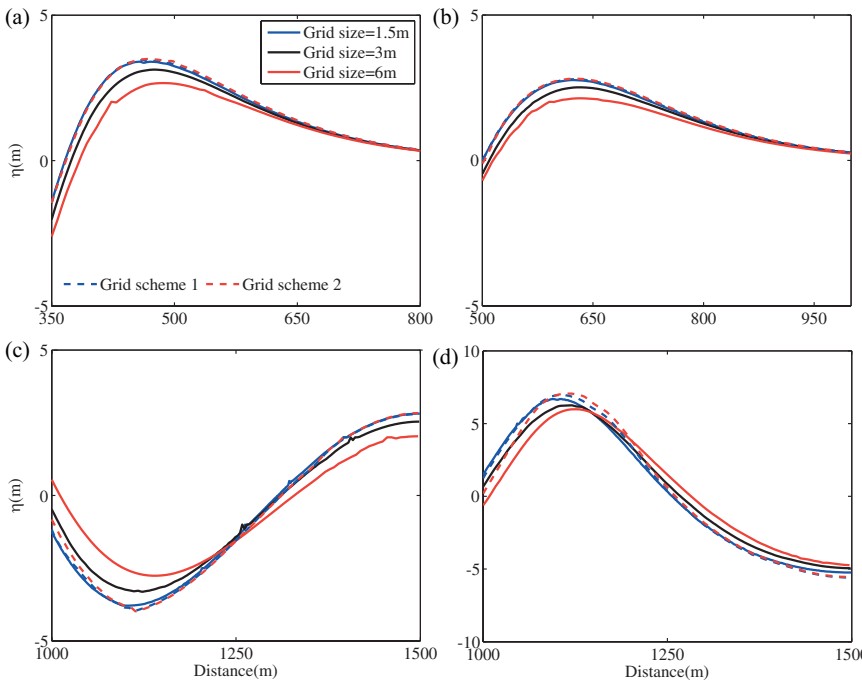

**Figure 8.** Computational results of different mesh schemes compared with that of original global uniform mesh calculation: (**a**) t = 15 s; (**b**) t = 20 s; (**c**) t = 50 s; and (**d**) t = 60 s.

**Table 5.** Mesh number and calculation time.

| Mesh Schemes | Mesh Number | Calculation Time (Hour) |
|---|---|---|
| 1.5 m global uniform mesh | 2,960,000 | 5.9 |
| Mesh scheme 1 | 960,000 | 1.45 |
| Mesh scheme 2 | 490,000 | 0.27 |

To further optimize efficiency, mesh scheme 2 is proposed to narrow the definition range of the NF. From the calculation results, the calculation under mesh scheme 2 still has good accuracy and the results at t = 15 s and t = 20 s are similar to those of mesh scheme 1 and 1.5-m fine global uniform mesh. As time progresses, there is a slight difference in the calculated wave phase (see Figure 8c,d), which is less accurate than mesh scheme 1. However, the total mesh number is 83% lower than the original fine mesh and 50% less than mesh scheme 1, the calculation speed is also significantly improved, which is 21.9 times that of uniform mesh and 5.4 times that of mesh scheme 1. Mesh scheme 2 is more ideal than mesh scheme 1 within the error tolerance. The proposed and calculated results of mesh scheme 2 show that the range of the NF is not fixed. Under certain conditions, the range of the fine mesh can be further reduced to improve computational efficiency.

## 5. Real-Word Application: Landslide Tsunami in Laxiwa Reservoir

In practical engineering, complex physical processes and large computing areas often present computational difficulties or slow calculations. The Guobu slope of the Laxiwa Reservoir in China has long been considered a potential landslide risk. Because of the particularity of the landslide location (near the dam), the height and intensity of the wave generated by a slide will endanger dam safety. Additionally, owing to the persistence of tsunami phenomena and a long wavelength in large-scale regions, the damage caused in the lower reaches of the river becomes a focus. The Laxiwa model has all the characteristics mentioned above, and the target region chosen contains both an NF and a PF, making it a suitable case for validating our mesh scheme. The 3-D geometric model (Figure 9b) was established according to the real-complex terrain (Figure 9a). Detailed parameters can be found in Table 6.

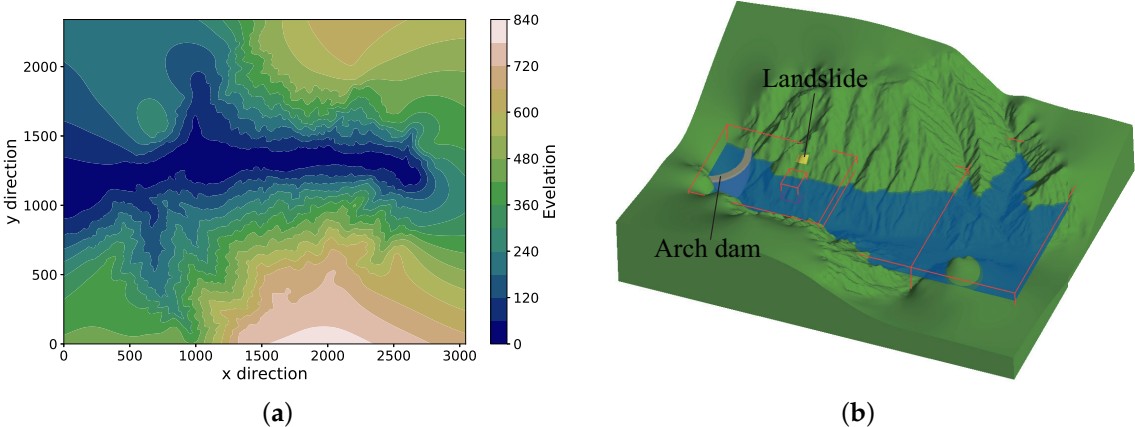

**Figure 9.** Large-scale numerical model of Laxiwa Reservoir, China. (**a**) Laxiwa terrain. (**b**) Geometric model.

**Table 6.** Parameters in the Laxiwa model.

| Parameter | Value |
|---|---|
| landslide type | rigid block |
| landslide volume | 86,960 m$^3$ |
| landslide density | 2650 kg/m$^3$ |
| dam height | 250 m |
| still water surface height | 230 m |
| water density ($\rho$) | 1000 kg/m$^3$ |
| calculation region in x,y,z direction | 3045 m $\times$ 2340 m $\times$ 805 m |

Based on the above discussion and the strategy of the mesh scheme, a conclusion can been reached regarding the accurate description of NF waves being key to improving the numerical simulation accuracy of the landslide tsunami. Thus, the mesh scheme, constituted by different mesh sizes, can be used in an engineering case. A smaller mesh block with minimum mesh size is applied to the NF around the landslide as shown in Figure 9b. The other blocks with larger mesh sizes are constructed into different shapes to fit the terrain and calculation domain. More detailed information is introduced in Figure 10.

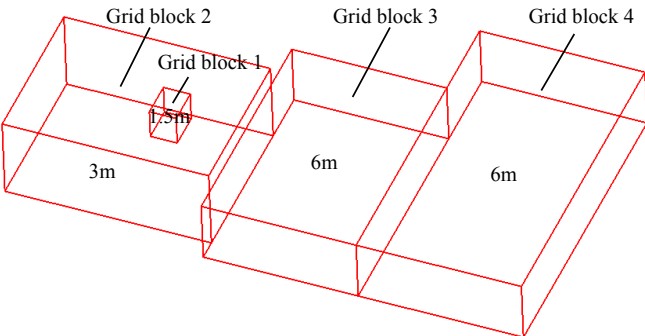

**Figure 10.** Mesh scheme application for engineering. The size of block 1: 130 m $\times$ 170 m $\times$ 150 m; the size of block 2: 1000 m $\times$ 750 m $\times$ 400 m; the size of block 3: 750 m $\times$ 1100 m $\times$ 300 m; the size of block 4: 800 m $\times$ 1500 m $\times$ 300 m.

The height of the initial impulsive surge caused by the landslide can often be used to evaluate the intensity of the landslide tsunami related to disaster prevention and prediction. For validation of the proposed mesh scheme in Laxiwa, two more global uniform mesh sizes of 3 and 6 m are used for calculations under the same conditions. Compared with the initial wave height calculated under various mesh schemes, the computational wave level decreases as the mesh size increases (as shown in Figure 11). The highest wave result presented in three solutions are 16.59, 19.58, 26.94 m. By comparing the dam-top elevation (dam height is 250 m, and still-water depth is 230 m), the wave level is 6.94 m higher than the dam top in the mesh scheme solution, whereas that of the mesh solution of 3 and 6 m are 0.42 and 3.41 m lower than the dam top, respectively. Although the location producing the highest wave is in block 1, the accuracy of the calculation in block 2 is considerably improved as block 1 is refined. The capture capacity under the mesh scheme increases by 37.59% compared with the 3 m global mesh. The breaking and splashing of waves can, thus, be captured. When the landslide is very close to the dam, the calculation result of this first impulsive wave plays a decisive role and may cause an overtopping process [8].

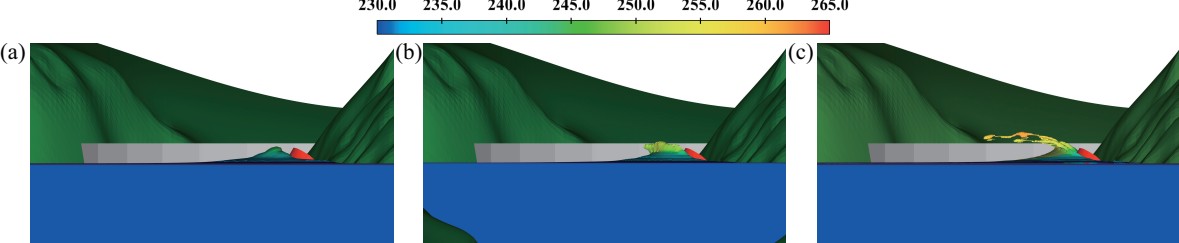

**Figure 11.** Highest initial wave in the near field at t = 10 s: (**a**) 6-m global uniform mesh; (**b**) 3-m global uniform mesh; and (**c**) mesh-scheme solution.

When the landslide stops moving and the wave enters the propagation stage, the different numerical result calculated in the NF will continue to appear in the PF (see Figure 12). Although the range of propagation is consistent, the wave height is different at 30 s. As the mesh size decreases, the calculation result shows a decrease in the intensity of the wave. However, the mesh-scheme rendering preserves the previous calculation accuracy and captures capabilities with the propagation-field coarse mesh. To monitor the wave height near the dam, a measuring point is placed to record the date and the results of time-dependent wave height under three solutions are depicted in Figure 13. The numerical difference first occurs when the lowest water level appears and remains afterwards. Because of wave propagation and energy attenuation, the capture capability of the mesh-scheme solution still has an advantage, but it is not as considerable as presented in the NF. The local refinement mesh scheme, considering NF and PF, has higher precision than the global coarse mesh under the same conditions.

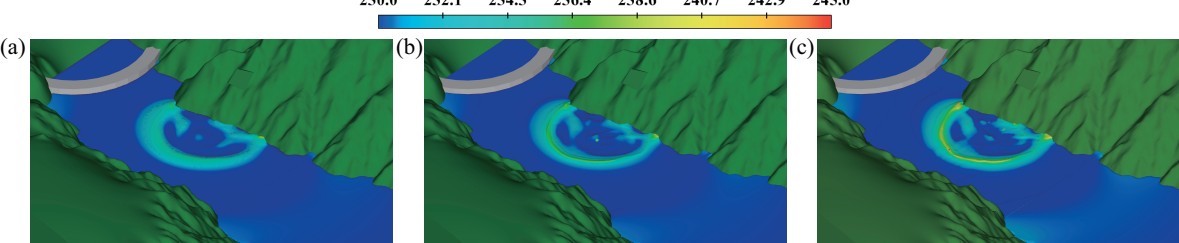

**Figure 12.** Wave height in the propagation field at t = 30 s: (**a**) 6-m global uniform mesh; (**b**) 3-m global uniform mesh; and (**c**) mesh scheme solution.

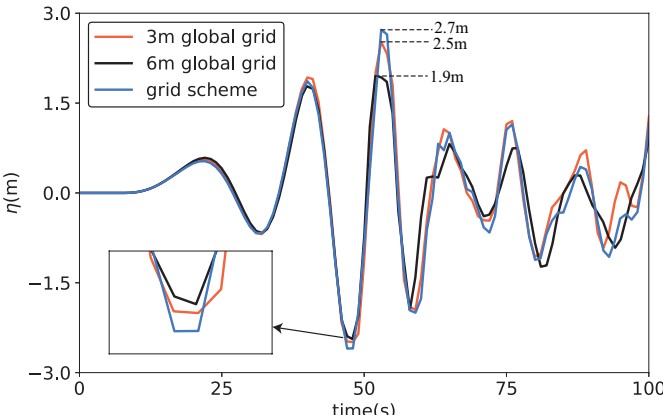

**Figure 13.** Wave height near the dam.

For the computational cost, the mesh scheme considerably reduces the mesh number (see Table 7) and improves the computational efficiency because fine mesh is only needed in the NF. The calculation speed is nonproportional to the mesh number because of the time it takes to write and store large

amounts of data. From a more extreme perspective, 1.5 m global-uniform mesh can be used for a more accurate result. However, the calculation and storage requirements are beyond the capabilities of most personal computers. With limited hardware capabilities, the special treatment in NF rather than global fine mesh can be considered a better solution for large-scale simulations. Compared to landslide-tsunami simulation in Reference [20] where the Flow-3D Code is adopted, the multi-size mesh blocks (while the single mesh block is used by Yin et al. [20]) are considered for a balance between efficiency and accuracy in this study. The sensitivity analyses of the mesh size and computational region are also conducted to serve the mesh-generation scheme. It is also worth noting that the hydraulic structure can also be considered in the model to make the simulation closer to reality. A high-speed and accurate calculation of the landslide-tsunami problem becomes feasible in personal computers.

**Table 7.** Mesh number and calculation time.

| Mesh Schemes | Mesh Number | Calculation Time (Hour) |
|---|---|---|
| 3-m global uniform mesh | 16,297,154 | 22.3 |
| 6-m global uniform mesh | 2,238,731 | 2.5 |
| Mesh scheme | 7,427,669 | 4.8 |

## 6. Conclusions

In this study, by comparing experimental data, it is shown that RANS and continuity equations can be used to accurately simulate the process of landslide tsunamis in both submarine and subaerial scenarios. Scale effects are investigated to establish the suitability of the model for real-world scale use. Multi-size mesh blocks are adopted to satisfy the specific calculation accuracy requirements of different regions. The 3-D numerical model can serve complex calculation conditions involving irregular topography, landslide surface, and hydraulic structures.

Based on GCI analysis, 14 cases of global uniform mesh were designed to investigate the dependence of the numerical results of the waves on computational domain dividing, and mesh size. The calculation area was divided into NF and PF according to the size of the mesh required to accurately capture the wave features. It can be concluded that the numerical result of the wave is highly dependent on the resolution of the mesh in the near field but less dependent on the mesh in the propagation field.

According to the analysis of mesh dependence, a mesh scheme was proposed and applied to the original numerical setting. The accuracy was considerably improved over the global original uniform coarse mesh, and the mesh number was significantly reduced. Two sets of new mesh schemes can increase calculation speed by 4.1 and 21.9 times. Thus, accurate simulation of NF is key to numerical simulation of landslide tsunamis. Finally, the mesh scheme was applied to the Laxiwa Reservoir in China. The mesh scheme achieved good results at an initial impulsive wave height, propagation wave intensity, and wave height near the dam. It considerably increased the speed of calculation. The research results provide guidance and reference values for rapid response and accurate calculation of actual engineering landslide-tsunami disasters. A more accurate definition of the NF range to obtain a better mesh solution is one of the key problems for a follow-up study.

**Author Contributions:** Conceptualization, G.L. and G.C.; methodology, G.L. and G.C.; software, G.C. and P.L.; validation, G.C. and P.L.; formal analysis, G.L., G.C. and H.J.; investigation, G.L. and G.C.; resources, G.L. and H.J.; data curation, G.C. and P.L.; writing–original draft preparation, G.C. and P.L.; writing–review and editing, G.L. and H.J.; supervision, G.L.; project administration, G.L.; funding acquisition, G.L. and H.J.

**Funding:** This research was funded by the Key R & D Projects in Shaanxi Province, China. Grant number 2018SF–352; National Natural Science Foundation of China. Grant number 51909211; and Natural Science Basic Research Plan In Shaanxi Province of China. Grant number 2019JQ-744.

**Acknowledgments:** We thank Shanshan Li for providing help to our final version. Sincere thanks also go to two anonymous reviewers and for their very helpful comments and suggestions, which have helped us clarify and improve the manuscript.

**Conflicts of Interest:** The authors declare no conflict of interest.

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
