# Peer review of "Efficient and Accurate 3-D Numerical Modelling of Landslide Tsunami"

_water, doi:10.3390/w11102033_

Round 1

Author Response

Dear Reviewer:

According to the comments from you, we have carefully revised our manuscript. The revisions have been clearly highlighted by red color in PDF file. Our responses to the your comments are written as follow.

Once again, thank you very much for your comments and suggestions.

Yours

Sincerely

Guodong Li, Guoding Chen, Pengfeng Li and Haixiao Jing

Comment: The authors have made considerable effort to discuss the advantage of using non-uniform grids. However, it seems to me that the use of unstructured mesh is standard. I am also puzzled by the results shown in Fig. 2 and 3, where two sets of simulations with different scaling are compared with the same measurements. Regarding the near-field and far-field results, I suspect that short impulsive waves are more significant in the region closer to the impact and therefore the coarse grids are capable of doing a reasonable job capturing the propagation of long waves.

Response: In this study, the mesh generation scheme is proposed based on the sensitivity analysis. Different from previous unstructured mesh, this study illustrates the error source during the landslide-tsunami simulation and emphasizes significance of accurate simulation in near-field. It is important to emphasize that the purpose of non-uniform is to refine the near-field region and determine the impact of near-field mesh refinement on the calculation results, rather than comparing the superiority of non-uniform meshes and unstructured meshes. For scale effect, the large-scale numerical results are minified according to the Froude similarity criterion and then compared to the indoor experimental data. According to the comments form reviewers, we have provided more discussion about scale effects by referring to relative literatures (Heller et al., line 58 & line 180).

Reviewer 2 Report

This paper presents a new numerical model for landslide tsunami based on the 3D Reynolds-averaged Navier stokes (RANS) equations, and a local mesh refinement grid scheme. The paper tests the accuracy and efficiency of the model and grid scheme. The model is validated using experimental data. Computational efficiency and the scale effect are investigated to establish the suitability of the model for simulating real-world scale scenarios.
The paper concludes that the model performs well using local mesh refinement for simulating a real scale landslide tsunami scenario.
They recommend that the definition of the near-field region needs further research in order to further optimise the mesh and simulations, because the mesh in the near-field has a strong influence on the numerical wave result.

I believe this is the first time this precise type of numerical model Flow-3D with RANS for a large scale 3D model of sub aerial landslide tsunami. But this needs to be pointed out, with discussion and explanation. How does this differ/novel from other studies that have used Flow-3D (e.g. Horillo, 2013) or other models that have used RANS (Ma, 2013)?
Has anyone in the past modelled this real-world scenario at all?

General comments:
The paper would benefit from additional discussion/explanation of some ideas.

• The test case is a submarine example, but real-world case looks sub aerial. I would suggest a sub aerial test case and before a sub aerial real-world scale example instead of in addition. Need to clarify the difference between sub-aerial and submarine slides and clarify which they are looking at in each case. Unless it can be justified why a submarine example is appropriate, this section should be replaced with a sub-aerial example.
• In the experimental test case they should also compare to other numerical model results, as many other studies have used this experiement.
• More discussion of scale effects needed, including referencing appropriate literature e.g. Heller.
• would benefit from a discussion and comparison of the pros and cons of this type of model (3D RANS) over the many other numerical models in the literature that have been used to simulate landslide-tsunami. Why is this better/novel?
• needs more discussion of the simplifications and their implications, e.g the deformation of the landslide are not considered, what are the implications of this, if any?
• slightly limited literature, there are more and better examples available for some of the points discussed

Specific comments by line number:
3. replace "disaster" with "disasters"
6. needs "to the" between 2according2 and 2Froude"
9. replace "mesh" with "meshes"
17. replace "landslide" with "landslides"
19. replace "the" with "a"
33. needs a definition of the scale effect, why it should be considered? what phenomena does it include? etc perhaps references to Heller et al 2008: Scale effects in sub-aerial landslide generated impulse waves and others?
38. remove "the", replaces "model" with "models". replace "has" with "have".
39. replace "the available model". not sure what this means?
41. this sentence needs restructuring so it doesn't start with "because2.
42. "2nd order 4th order" should say "2D [two-dimensional] 4th order"
48. "uniform mesh size generally adopted", but there are examples of where there hasn't been the case, and these should be discussed e.g.. Hill et all 2014 https://www.sciencedirect.com/science/article/pii/S1463500314001164
and there are other tsunami studies that have used nested grids, these should be references.
53. "In this study, more attention is paid" should be something like "this study focuses on"
59. conclusion of what? efficiency of model? accuracy of model? suitability of model?
64. Flow-3D, what about the previous slide tsunami studies that used Flow-3D such as Horillo et al 2013 https://agupubs.onlinelibrary.wiley.com/doi/full/10.1002/2012JC008689
How does this model set up differ from that to make it novel? This is should be identified and explained.
77. what might be the consequences of ignoring the deformation of the landslide and collision with the slope? Why is this justified? What effects might not be being captured? are these important?
86. these references are all recent, what about Assier-Rzadkiewicz
et al. (1997) http://ascelibrary.org/doi/abs/10.1061/(ASCE)0733-950X(1997)123:4(149)
94. an image of this mesh of orthogonal hexagons would be beneficial
116. can this conclusion be drawn from just these results? Is there anything else that has yet to be investigated/considered?
This Heinrich experiment has been modelled using other models Assier-Rzadkiewicz
et al. (1997) http://ascelibrary.org/doi/abs/10.1061/(ASCE)0733-950X(1997)123:4(149) Ma 2013 (https://www.sciencedirect.com/science/article/abs/pii/S1463500313001170), Smith et al 2016 (https://www.sciencedirect.com/science/article/pii/S1463500316000354)
Capone 2010 used SPH: https://www.tandfonline.com/doi/abs/10.1080/00221686.2010.9641248
There may well be others.
Ma et al 2013 also use K-E RANS turbulence model (but for the slide) (Lin and Liu, 1998a,b; Ma et al., 2011, 2013).
Ma, G. 2011. http://onlinelibrary.wiley.com/doi/10.1029/2010JC006667/abstract
Lin, Liu, 1998b.: http://onlinelibrary.wiley.com/doi/10.1029/98JC01360/abstract
Can plots be made comparing the current model to these previous models? and/or comparisons in terms of computational efficiency e.g. number of mesh nodes, cpu time. Is this model better than all the others that have modelled the same experiment? is it faster? is it more accurate?
131. can they provide a reference for the GCI?
Table 3. "Number of grids", "number of mesh", should this be "number of grid cells in the mesh"?
lines - 198-191: can this sentence be made clearer, do you mean inaccurate capturing of the wave characteristics in the near-field?
191. - what about mesh adaptivity e.g. smith et al 2016 https://www.sciencedirect.com/science/article/pii/S1463500316000354#bib0052 is this not also a method of local mesh refinement?
263. last sentence needs rewording/explaining better. Do you mean "the numerical result of the wave his highly dependent on the resolution of the mesh in the near-field, but less dependent on the mesh in the propagation field"?

Author Response

Dear Reviewer:

According to the comments from you, we have carefully revised our manuscript. The revisions have been clearly highlighted by red color in PDF file. Our responses to the your comments are written as follow.

Once again, thank you very much for your comments and suggestions.

Yours

Sincerely

Guodong Li, Guoding Chen, Pengfeng Li and Haixiao Jing

Comment 1: The test case is a submarine example, but real-world case looks sub aerial. I would suggest a sub aerial test case and before a sub aerial real-world scale example instead of in addition. Need to clarify the difference between sub-aerial and submarine slides and clarify which they are looking at in each case. Unless it can be justified why a submarine example is appropriate, this section should be replaced with a sub-aerial example.

Response: We have added a sub aerial test case in Section 3. In addition, we prefer to keep the submarine test and thus the model can be verified by both subaerial and submarine cases.

Comment 2: In the experimental test case they should also compare to other numerical model results, as many other studies have used this experiment.

Response: According to the comment, other numerical results (Nasa-vof2D results of Heinrich, ISPH results of Yeylaghi et al., DEM-SPH results of Tan et al., DualSHPysics results of Heller et al.) have been added to compare our result. (see in Section 3)

Comment 3: More discussion of scale effects needed, including referencing appropriate literature e.g. Heller.

Response: We have supplied the more discussion of scale effect in introduction (line 58) and Section 3 (line 180). More literatures (e.g. Heller et al. ) have been referenced.

Comment 4: would benefit from a discussion and comparison of the pros and cons of this type of model (3D RANS) over the many other numerical models in the literature that have been used to simulate landslide-tsunami. Why is this better/novel?

Response: We have supplied the 3D RANS cases in introduction (line 50), and the pros and cons of this type of model are supplied in Section 2. The comparison is conducted in Section 5 and novelty of the proposed model is also supplied (line 329). The multi-size mes blocks(while the single mesh block is used by Yin et al. ) are considered for a balance between efficiency and accuracy in this study. The sensitivity analyses of the mesh size and computational region are also conducted to serve for mesh generation scheme. It is also worth noting that the hydraulic structure can also considered in model to make the simulation closer to reality.

Comment 5: needs more discussion of the simplifications and their implications, e.g the deformation of the landslide are not considered, what are the implications of this, if any?

Response: According to the comment, we have discussed the simplifications and implication in Section 2 (line 122). As reported, rigid slides produce more turbulence as well as greater wave runup and wave height. The bluff body effect of rigid landslides seems to significantly influence the wave height and runup (Kaushik et al, https://www.onepetro.org/conference-paper/OTC-20293-MS). In our work, the mesh generation schemes and efficient and accurate numerical model establishment are focused. It needs to be emphasized that the landslide block deformation and fragmentation are not considered and the block itself is treated as a rigid body. In addition, neglecting the deformation is consistent with the verification experiment selected in the following sections.

Comment 6: slightly limited literature, there are more and better examples available for some of the points discussed.

Response: We have carefully rechecked the full text and corrected the examples and references.

Comment 7: replace "disaster" with "disasters"

Response: We have corrected.

Comment 8: needs "to the" between 2according2 and 2Froude"

Response: We have corrected.

Comment 9: replace "mesh" with "meshes"

Response: We have corrected.

Comment 10: replace "landslide" with "landslides"

Response: We have corrected.

Comment 11: replace "the" with "a"

Response: We have corrected.

Comment 12: needs a definition of the scale effect, why it should be considered? what phenomena does it include? etc perhaps references to Heller et al 2008: Scale effects in sub-aerial landslide generated impulse waves and others?

Response: We have supplied the more discussion of scale effect in introduction (line 58) and Section 3 (line 180). More literatures (e.g. Heller et al. ) have been referenced.

Comment 13: remove "the", replaces "model" with "models". replace "has" with "have".

Response: We have corrected.

Comment 14: replace "the available model". not sure what this means?

Response: We have corrected.

Comment 15: this sentence needs restructuring so it doesn't start with "because2.

Response: We have corrected.

Comment 16: "2nd order 4th order" should say "2D [two-dimensional] 4th order"

Response: We have corrected.

Comment 17: "uniform mesh size generally adopted", but there are examples of where there hasn't been the case, and these should be discussed e.g.. Hill et all 2014 https://www.sciencedirect.com/science/article/pii/S1463500314001164

Response: We have corrected the sentence and supplied the new references.

Comment 18: "In this study, more attention is paid" should be something like "this study focuses on"

Response: We have corrected.

Comment 19: conclusion of what? efficiency of model? accuracy of model? suitability of model?

Response: We have revised the conclusion in last section.

Comment 20: Flow-3D, what about the previous slide tsunami studies that used Flow-3D such as Horillo et al 2013 https://agupubs.onlinelibrary.wiley.com/doi/full/10.1002/2012JC008689 How does this model set up differ from that to make it novel? This is should be identified and explained.

Response: We have supplied the 3D RANS cases in introduction (line 50) . The characteristics and details of the proposed model are supplied in Section 2. The comparison is conducted in Section 5 and novelty of the proposed model is also supplied (line 329). We have also pointed out the novelty of the model in conclusion.

Comment 21: what might be the consequences of ignoring the deformation of the landslide and collision with the slope? Why is this justified? What effects might not be being captured? are these important?

Response: According to the comment, we have discussed the simplifications and implication in Section 2. (line 122).

Comment 22: these references are all recent, what about Assier-Rzadkiewicz et al. (1997) http://ascelibrary.org/doi/abs/10.1061/(ASCE)0733-950X(1997)123:4(149)

Response: We have supplied the reference.

Comment 23: an image of this mesh of orthogonal hexagons would be beneficial

Response: In this paper, the mesh scheme involves multi-size mesh blocks. In each block, the mesh size is uniform. For clearly, the mesh blocks are expressed by figure (e.g. Figure 10), and the mesh size is illustrated in words.

Comment 24: can this conclusion be drawn from just these results? Is there anything else that has yet to be investigated/considered?

Response: According to the general comments, we have revised the validation, method description, scale effect discussion, etc. The conclusion have been revised and can be better supported.

Comment 25: can they provide a reference for the GCI?

Response: We have supplied the references. (line 207)

Comment 26: Table 3. "Number of grids", "number of mesh", should this be "number of grid cells in the mesh"?

Response: We have corrected.

Comment 27: lines - 198-191: can this sentence be made clearer, do you mean inaccurate capturing of the wave characteristics in the near-field?

Response: We have corrected the sentence.

Comment 28: what about mesh adaptivity e.g. smith et al 2016 https://www.sciencedirect.com/science/article/pii/S1463500316000354#bib0052 is this not also a method of local mesh refinement?

Response: Smith et al. use mesh adaptively to capture the deformation of the landslide. In this study, the rigid block is used for simplification of model. In addition, it has been verified that the shape of the landslide has a slight influence in far field. (Lo at al. 2017 https://www.cambridge.org/core/product/identifier/S0022112017002518/type/journal_article). Moreover, duo to the deep water depth in reservoir, the landslide has less effect on free surface flow in its later movement. Thus, the multi-size mesh blocks are adopted in this paper.

Comment 29: last sentence needs rewording/explaining better. Do you mean "the numerical result of the wave his highly dependent on the resolution of the mesh in the near-field, but less dependent on the mesh in the propagation field"?

Response: We have corrected the sentence.